# A Microscale Analysis of Thermal Residual Stresses in Composites with Different Ply Orientations

**DOI:** 10.3390/ma16196567

**Published:** 2023-10-06

**Authors:** Yanfeng Wang, Qi Wu

**Affiliations:** 1State Key Laboratory of Mechanics and Control for Aerospace Structures, Nanjing University of Aeronautics and Astronautics, Yudao Street 29, Nanjing 210016, China; 2Institute of Industrial Science, The University of Tokyo, 4-6-1 Komaba, Meguro-ku, Tokyo 153-8505, Japan

**Keywords:** thermal residual stresses, ply orientation, microscale model, composites, high-performance computing

## Abstract

Composites, such as fiber-reinforced plastics, are produced using layering prepregs with varying ply orientations to achieve enhanced mechanical properties. However, this results in intricate residual stresses, which are influenced by the forming process and ply orientation. In this study, three representative microscopic models—featuring discrete fiber and resin—represent unidirectional, cross-ply, and angle-ply laminates. These models underwent simulations under three different cooling histories using the finite element method. The findings suggest that ply orientation does not significantly influence temperature distribution. However, it significantly impacts the von Mises stress in the fiber closest to the interface between two stacked laminae. This differs from the inter-laminar stresses determined with the macroscopic lamination model. Apart from the free edge, which exhibits a complex stress distribution, the von Mises stress within a unit cell displays a recurring pattern. The magnitude of the von Mises stress decreases as the ply orientation angle increases and shifts when a temperature gradient is present throughout the composite’s thickness. This study provides valuable insights into the mechanics of residual stresses at the microscopic level and highlights potential defect areas influenced by these stresses.

## 1. Introduction

A composite lamina exhibits transverse isotropy, with high strength along the fiber direction and comparatively lower strength transverse to the fiber direction. By stacking multiple laminae with specific ply orientations, the mechanical performance of a composite laminate can be substantially enhanced. For instance, engineers often utilize ply orientations of 0, ±45, and 90° to optimize aircraft performance [1]. Modern tape placement technology has also enabled stacking laminae with alternate orientations [2]. However, this process does not adequately enhance interlaminar toughness [3]. Moreover, forming-induced thermal residual stresses, which are determined with the composite arrangement and forming temperature, affect the fracture toughness of composite laminates [4]. Therefore, examining the impact of ply orientation and temperature on thermal residual stress is relevant and necessary.

Stango and Wang explored the thermal residual stresses in thermosetting composites using a quasi-3D finite element (FE) method. They highlighted how interlaminar stresses distributed along ply interfaces were influenced by the laminate stacking sequence [5]. Wang and Sun employed laminated plate theory to study the residual stresses in thermoplastic composites. They underscored the significant impact of ply orientations, such as [45/−45], on residual stresses [6]. Although other macroscale simulations have been performed on residual stresses [7,8], these traditional approaches, which characterize a lamina as transversely isotropic material, overlook the composite’s microscopic structure of fiber and resin. This leads to stress evaluation errors because microscopic stress is often several times higher than macroscopic stress [9]. Besides macroscale FE simulations, multiple microscale studies on residual stresses have been conducted [10,11,12]. However, these do not effectively probe the interlaminar residual stresses that influence composite mechanical properties. This is due to the fact that a microscopic model containing only one or a few unit cells cannot depict the residual stresses shaped by ply orientation and cooling histories [13]. Thus, a comprehensive study of the forming-induced residual stresses in a composite laminate using an FE simulation strategy, which can represent the mechanical change in the composite interlaminate during its forming process, is preferred. The FE model should be sufficiently fine to the microscale level for effectively separating the fiber and resin, while it should be sufficiently large to enclose the ply orientation as well as the temperature gradient across the composite thickness.

Whitcomb et al. devised a microscale composite model that integrated numerous discrete fibers and resins. They utilized a high-performance, in-house FE software capable of performing linear elastic analyses on a supercomputer system to simulate microscale stresses when the composite laminate undergoes tensile loading [14]. Their findings showed that the interlaminar normal stress near the free edge significantly deviates from the stress found deeper within the laminate. This discrepancy also exists in comparison to theoretical predictions. This is because the continuous resin distribution near the boundary of two plies does not result in a stress singularity present in the bi-laminar model. Later, in 2019, they conducted an FE simulation on a quasi-isotropic laminate under tensile loading. However, their discussions predominantly focused on interlaminar stresses near the free edge and between [90] and [0] plies [15]. In the same timeframe, our team formulated a macro-micro simulation strategy, along with another proprietary FE software, to explore thermal residual stresses in composites [16]. Our method compares macroscopic and microscopic residual stresses, showcasing actual values when the composite laminate under examination is distant from the free edge. In essence, our approach does not simulate the residual stresses proximal to the free edge, an area susceptible to damage. Therefore, to our understanding, a comprehensive and in-depth exploration of microscopic thermal residual stresses in various regions of a composite laminate, particularly in relation to ply orientation and cooling history, remains elusive.

Given the significance of residual stresses, we employed simulations on three representative FE models with three typical ply orientations: unidirectional [0], angle-ply [0/45], and cross-ply [0/90] laminates, under two temperature profiles. These models, with numerous degrees of freedom (DoFs), were processed using a high-performance computing infrastructure combined with a newly developed program boasting extensive parallel computing capability. The outcomes from this study not only deepen our comprehension of microscopic residual stress distribution throughout a composite laminate but also shed light on the influences of ply orientation and temperature.

## 2. Simulation Configuration

### 2.1. Forming of Thermoplastic Composite

Although the dynamic process of thermoplastic composite formation is consistently thermomechanical, different models should be employed depending on the composite system in use. This study utilized formulas tailored for composites made of carbon fiber (HexTow AS4 carbon fiber, manufactured by Hexcel Corporation, Stamford, CT, USA) and amorphous thermoplastics such as polyetherimide (PEI, manufactured by Mitsubishi Plastic, Inc. Tokyo, Japan). During heat transfer in a carbon-fiber/PEI-matrix composite laminate, the PEI experiences a much larger volume shrinkage when compared to the carbon fiber. This discrepancy in the coefficients of thermal expansion (CTE) between the fiber and resin leads to microscopic thermal residual stresses. Concurrently, the arrangement and cooling history introduce macroscopic thermal stresses, as different laminae exhibit varied shrinkages. However, both types of stress tend to diminish over time due to the viscoelastic nature of the resin.

Table 1 presents the properties of the carbon fiber. This fiber is transversely isotropic, with thermal and mechanical parameters that remain consistent irrespective of temperature and time [17]. Given these characteristics, a basic elastic model is employed to represent the fiber, as it typically remains intact during the forming process. In contrast, PEI is isotropic, exhibiting temperature-dependent attributes such as heat capacity, conductivity, volume, and thermo-viscoelasticity, as previously determined with our measurements [16]. The recorded raw properties are denoted with open dots in Figure 1. Subsequently, piecewise linear functions capture the heat capacity and conductivity, the empirical Tait equation represents volume, and the Maxwell model—expressed in a Prony series format—captures the linear thermo-viscoelasticity for any specified time and temperature when combined with a shift factor [18]. All corresponding fitting curves are illustrated as solid lines in Figure 1, and the respective mathematical models can be found in the Appendix A.

### 2.2. Simplified Geometric Model

Figure 2 depicts three representative models, each consisting of two laminae. Both the top and bottom laminae are aligned with the x-z plane. In the bottom lamina, fibers are oriented parallel to the z-axis. Conversely, the top lamina is rotated by 0, 45, and 90° relative to the y-axis for the unidirectional [0], angle-ply [0/45], and cross-ply [0/90] laminates, respectively. The hatched portions in Figure 2b–d provide transparent views of the respective models, revealing the arrangement of fibers within the matrix. These configurations encapsulate the typical layouts of composites because industrial composite laminates often consist of combinations derived from these three models. For ease of subsequent discussions, two planes are identified: the interlaminar plane, situated between the top and bottom laminae, and the fiber plane, representing the cross-section of the fiber closest to the interlaminar plane within the bottom lamina, as illustrated in Figure 2.

All models measure 200 × 200 × 60 μm (L × W × H). Simulating a composite made of discrete fiber and resin with real dimensions poses a challenge even for high-performance clusters or supercomputers due to the vast number of DoFs. These representative FE models, having a length over three times their thickness, can yield meaningful results near and far from the composite laminate’s free edge with an affordable computational cost. This efficiency will be further demonstrated in Section 3.2. Carbon fibers, each with a diameter of 7 μm, are uniformly distributed in a square arrangement throughout the composite. This distribution corresponds to a fiber volume fraction of 38.5%, as the local non-uniformity of carbon fiber does not affect the distribution of residual stress [14]. Given the minimal relative shrinkage between the carbon fiber and its surrounding matrix, it is assumed that the fibers are firmly bonded to the matrix. This implies that no friction or sliding takes place during the forming process.

### 2.3. Boundary Condition

In Figure 3, the temperature profile is illustrated with a red line, exhibiting a decline from 240 °C at 9 s to 20 °C at a constant rate of 20 °C/s. This profile is indicative of water cooling. Even though the peak molding temperature can reach up to 350 °C, existing research indicates that minimal residual stress is accumulated above 240 °C due to PEI’s low modulus and short relaxation time [16]. The blue line depicts a temperature profile that cools at a rate of 2 °C/s, which is ten-fold slower and represents ambient cooling. This linear decrease in temperature ideally captures the transition of PEI from its rubbery to glassy stage, considering the glass transition temperature of PEI is roughly 210 °C. With these cooling histories in mind, we set three distinct thermal boundary conditions to probe the impact of temperature on residual stresses, assuming heat transfer is exclusive to the model’s thickness direction. Firstly, the entire FE model undergoes rapid cooling, guided by the red profile, aiming to discern the basic dynamics of thermal residual stresses while eliminating the effects of the temperature gradient. Secondly, only the top surface follows the rapid-cooling profile. This simulation attempts to emulate real-world outside-to-inside cooling, or one-sided cooling, which naturally establishes a temperature gradient across the thickness. Finally, two Dirichlet boundaries—guided by the red and blue profiles in Figure 3—are applied to the model’s top and bottom surfaces, respectively. This approach strives to recreate a cooling process with a pronounced temperature gradient, similar to that observed in water cooling.

The boundary conditions for a stress analysis are intricate. Although a periodic boundary is commonly utilized for items exhibiting a periodic structure, it might be impractical when external loading is not uniform, as observed in the composite forming process with a temperature gradient. Furthermore, a periodic boundary cannot capture the complex stress distribution near the model’s edge. The prescribed boundary, suggested in the macro-micro simulation approach by interpreting results from the macro model, is highly dependent on homogenization accuracy. Most critically, neither the periodic nor the prescribed boundary can depict the stress distribution near a composite’s free edge. Drawing from the setup in reference [14], a comparable boundary was applied to the unidirectional and cross-ply laminate models. Given the inherent asymmetry, y-axis displacement on the bottom surface was restricted. Additionally, normal movements on the back and left surfaces are confined, as illustrated in Figure 4a. For the angle-ply laminate, the bottom surface’s y-axis displacement remained constrained. However, due to the absence of axisymmetry and the shifting angles of 0-degree fibers and angled fibers during thermoforming, one model corner was immobilized. Simultaneously, an opposite corner was tethered to a grounding spring to avert rigid body motion. The model’s remaining surfaces are left unconstrained, as depicted in Figure 4b.

### 2.4. Finite Element Analysis

The three models outlined in Section 2.2 were meshed utilizing 10-node tetrahedral elements. Considering the balance between computation time and precision, the fiber’s cross-section at the bottom lamina is represented with 18 elements, as illustrated in the inset of Figure 2. Hence, the cross-ply model comprised approximately 2.4 million nodes and 1.7 million elements. The other two models had comparable numbers of DoFs. The timestep for the dynamic thermomechanical analysis was set at 1 s. The adequacy and accuracy of this mesh size and timestep have been previously validated [19].

A software platform called “FrontCOMP_TP” was employed. The software is designed to manage up to 100 million DoFs and operate on a high-performance computing system. This software is programmed using weak coupling of a heat transfer analysis and mechanical analysis. To validate the software’s efficiency and trustworthiness of the material property models utilized, three tests were conducted, and their outcomes were compared with simulation results. In the first test, a PEI cantilever experienced continuous deflection under gravitational force due to a reduction in its rigidity as the ambient temperature increased. The simulation projected a deflection of 3.8 mm, aligning closely with the observed 3.5 mm deflection [19]. In the second test, a [90]8 PEI-matrix composite beam with a fiber volume fraction of 45% was subjected to rapid one-sided cooling after heating, resulting in a warpage. The simulated warpage was 1.45 mm, closely mirroring the actual measured warpage of 1.12 mm [16]. In the third test, the thermoforming process of a PEI-matrix composite laminate, measuring 100 × 100 × 3.14 mm (L × W × H), was simulated under a one-sided cooling condition while considering the tool-part interaction [20]. The simulated warpage was 1.44 mm, which was nearly identical to the actual recorded warpage of 1.55 mm. These results underscore the reliability of the FE simulations and software’s robustness.

By leveraging a computer cluster including 576 CPU cores (Xeon E5-2660/2.20 GHz), the software platform was used to calculate the large-DoF and 250-step (1 s = 1 step) models in Figure 2, resulting in a typical solving time of 1 h. To cooperate with the FE solver, pre- and post-processing software used was FEMAP (v11.2.1) and MicroAVS (v18.0), respectively.

## 3. Results and Discussions

### 3.1. Deformation of Model

Figure 5a,b display the deformation observed in the unidirectional and cross-ply composite models when both are cooled uniformly. A scale factor of 10 was applied for clarity in visual representation. In the unidirectional model, there is pronounced shrinkage in the direction transverse to the fiber direction, while minimal changes are observed along the fiber direction. This distinction arises due to the unidirectional lamina’s characteristics: it possesses a smaller CTE and larger modulus in the fiber’s direction, and conversely, a larger CTE and smaller modulus in the transverse direction. Similarly, the cross-ply and angle-ply composites demonstrate more pronounced shrinkage transverse to the fiber than along its direction. However, when the two laminae are at an angle to each other, they influence each other’s behavior. For instance, in the cross-ply laminate scenario, the top lamina compresses the one beneath it in the x-direction. Hence, the stress patterns near the interlaminar plane become intricate and are influenced by the orientations of the two interacting laminae.

### 3.2. Edge Effect

Figure 5a,b also illustrate the von Mises stress, denoted as *σ_Mises_*, on the cross-sections of x–y and y–z planes for the unidirectional and cross-ply laminates, respectively, after uniform cooling. For the unidirectional and cross-ply laminates, the stress distribution within the fibers near the free edge is intricate. However, a recurring stress distribution pattern is evident across the thickness along the x- or z-axis when fibers are distanced from the free edge. This edge effect also exists in the angle-ply laminate and is a pervasive phenomenon in plate-like structures where the length and width substantially exceed their thickness. A similar trend is observed in microscale models with randomly dispersed fibers, as noted in reference [14], and in a benchmark microscale model [16]. Relying on the empirical formulas provided in reference [14], it is determined that the extent of the edge effect is approximately the model’s thickness, which is 60 μm in this study. This estimation, combined with the varying stress distribution patterns observed near and distant from the free edge, affirms that the model size employed aptly differentiates the edge effect.

Figure 6 illustrates *σ_Mises_* near and distant from the free edge of the unidirectional and cross-ply laminates, as magnified from select positions in Figure 5. The intricate distribution of *σ_Mises_* near the free edge along the z-axis in the unidirectional laminate arises from the mismatch in the CTEs and moduli between the fiber and resin. The fiber, characterized with a larger Young’s modulus and smaller CTE, impedes the shrinkage of the resin, which possesses a larger CTE and smaller Young’s modulus. Guided by the shear lag principle, the fiber and resin exhibit consistent shrinkage with a temperature decrease when the z-coordinate is distant from the free edge, resulting in a recurring stress pattern. Within the same unidirectional laminate, the stress near the free edge along the x-axis and on the top surface registers a marginally larger magnitude, potentially attributed to boundary effects. The free top and right surfaces allow for greater resin shrinkage when compared to the bottom and left surfaces, where normal displacements are constrained [14]. Analogous to the unidirectional laminate, the cross-ply laminate also displays an edge effect near the free edges along the z- and x-axes. This pronounced edge effect primarily stems from ply orientation, although fiber–resin mismatches and boundary influences also play roles. However, the intricate stress distribution is found to be recurrent, also governed by the shear lag principle. The ensuing anisotropic shrinkage and edge effects can also be observed in the strain contour, which is not detailed here to maintain brevity.

### 3.3. Interlaminar Stresses

Beyond the edge effect, a distinct disparity between Figure 5a,b is evident in the stress distribution across the top and bottom laminae. In Figure 5a, *σ_Mises_* within the fiber exhibits minor fluctuations throughout its thickness. Conversely, in Figure 5b, *σ_Mises_* within the fiber sharply peaks at the bottom lamina. This underscores that such a divergence stems exclusively from the ply orientation, given that both models experience the same cooling process and share identical boundary conditions in the analysis.

In the classical bi-laminal theory [16], significant thermal interlaminar residual stresses exist in a cross-ply laminate because of the transversely isotropic properties, such as CTE, of a composite lamina. Unlike a unidirectional laminate that exhibits identical shrinkage throughout its thickness, the top and bottom laminae of a cross-ply laminate interact at the interface. Herein, a simplified example obtained using the fiber and resin properties at room temperature is presented for comparison. Using the Hashin homogenization method, the evaluated CTEs in the transverse and axial directions are 1.97 × 10^−5^ °C^−1^ and 9.44 × 10^−8^ °C^−1^, respectively. Hence, the [90] ply obstructs the shrinkage of the [0] ply, which forms a maximal *σ_Mises_* of 114.126 MPa near the edge of the bi-laminal cross-ply model. This value is much higher than the maximal *σ_Mises_* in Figure 5 and Figure 6. Therefore, any angle-ply laminate at a macroscopic scale will form interlaminar stresses when using the classical bi-laminal theory [16].

Figure 7a,b illustrate *σ_Mises_* on the interlaminar and fiber planes, respectively, observed from the bottom perspective at the model’s corner. A significant portion of *σ_Mises_* presented on the interlaminar plane is relatively minor. This discrepancy from the macroscale theoretical prediction marks the primary distinction between macroscopic and microscopic simulations. In the microscopic model, the top and bottom laminae are fused with the same resin, which possesses an identical CTE and Young’s modulus. This contrasts with the macroscopic model, where two adjacent laminates exhibit varying properties. Furthermore, there are sporadic light-yellow lines in the unidirectional laminate and red dots in the angle-ply and cross-ply laminates, marking areas where fibers from the top and bottom are in proximity. This observation also diverges from the conventional macroscopic forecast, which predicts a smooth transition across coordinates.

The substantial interlaminar residual *σ_Mises_* that appears to be absent in Figure 7a is redistributed to the fiber plane, as shown in Figure 7b. Overlooking the stress near the free edge and then comparing Figure 5 and Figure 7, it becomes evident that the thermal residual *σ_Mises_* is pronounced in the fiber plane nearest to the top lamina. This behavior suggests that the fiber plane serves as the boundary where the two laminae, each with different ply orientations, interact. The shrinkage in the top lamina exerts a force on the bottom lamina through the fiber plane, primarily as a shear force, particularly considering the significant difference in CTE between the fiber plane and resin. Additionally, blue lines manifest in areas with a higher resin concentration in Figure 7b, a pattern similar to what is observed in the unidirectional laminate depicted in Figure 7a.

### 3.4. Influence of Ply Orientation

The edge effect influences the stress distribution only within a confined region. Therefore, examining the recurring stress patterns distant from the edge is especially significant for large-sized composite laminates with minimal thickness. Figure 8 depicts the stress within a unit cell in the fiber plane, which is noticeably affected by ply orientation. Irrespective of ply orientation, all shear stresses tend towards zero, making the primary stresses the normal ones. In the fiber direction, the fiber experiences compressive *σ_zz_*, while, in contrast, the surrounding resin exhibits tensile *σ_zz_*. The distributions of *σ_xx_* and *σ_yy_* are intricate, but their absolute values are much smaller than the prevalent *σ_zz_*. Nevertheless, as the ply orientation angle increases, all normal stresses intensify in magnitude. Consequently, the corresponding *σ_Mises_* illustrated in Figure 8b is also influenced by ply orientation. This observation suggests that while ply orientation considerably affects normal stresses, it has a minor effect on shear stresses. The findings further suggest that substantial interlaminar stresses at the fiber plane can be effectively mitigated by adeptly designing neighboring plies with reduced orientation angles.

### 3.5. Influence of Cooling

The results discussed earlier were derived when employing the first type of thermal boundary, where the entire model cools uniformly. In the second scenario, with the Dirichlet boundary applied solely to the top surface, the temperature variances between the top and bottom surfaces of all three representative microscopic models were observed to be approximately 0.1 °C at 25 s. This subtle temperature gradient arises from the model’s micrometric thickness and minimal influence of ply orientation. Aspects such as density, specific heat capacity, and thermal conductivity remain consistent across the composite’s thickness, which is perpendicular to the fiber.

Contrasting the first and second temperature boundaries, which scarcely influence the thermal residual stresses in a microscopic model, the third boundary introduces a considerable temperature gradient. This results in intricate *σ_Mises_* patterns on the cross-sections of the unidirectional and cross-ply microscopic models, as illustrated in Figure 9. An edge effect is evident in both scenarios, with a similar breadth equating to the model’s thickness. Broadly, the magnitude range of *σ_Mises_* surpasses that presented in Figure 5. For instance, under the pronounced temperature gradient, the peak *σ_Mises_* in the fiber is approximately 1.2 times higher than under uniform cooling. This is consistent for unidirectional and cross-ply models.

In Figure 9a, disregarding the edge effect, *σ_Mises_* across the thickness predominantly follows a parabolic profile. Hence, the complete *σ_Mises_* distribution across the thickness results from the superimposed effects of the parabolic macroscopic stress from a bi-laminar model and the intricate microscopic strain from a unit cell. As shown in Figure 9b, the *σ_Mises_* in the top lamina resembles the patterns observed in Figure 5b. The *σ_Mises_* in the bottom lamina mirrors the patterns in Figure 5b but with amplified magnitudes. In terms of the cross-ply laminate, the cooling effect appears less pronounced, mainly because the impact of ply orientation substantially overshadows the effects of cooling.

## 4. Conclusions

In this study, thermal residual stresses in representative microscopic composite models, featuring three different layouts and cooling histories, were analyzed using a newly devised FE program. The findings revealed distinct stress patterns near and far from the free edge. Far from the edge, *σ_Mises_* exhibited repetitive patterns through the thickness in in-plane directions. The ply orientation played a pivotal role in shaping *σ_Mises_* distributions. Contrary to macroscopic model simulations, *σ_Mises_* within the resin near the interlaminar plane exhibited a gradual shift as opposed to abrupt changes. In contrast, the interlaminar stress manifested predominantly in the fiber closest to the interlaminar region of the bottom lamina. Notably, the variations in *σ_Mises_* were predominantly attributed to normal stresses rather than shear stresses. As the angle of ply orientation increased, *σ_Mises_* also increased, suggesting a higher probability of inter-delamination or other defects between laminae with wide ply orientation angles. To enhance the strength of composites, layer-by-layer stacking of laminae, each with minor orientation shifts, is more beneficial than adopting a cross-ply configuration. Besides ply orientation, cooling histories also influenced *σ_Mises_*, particularly when there was a pronounced temperature gradient across the thickness. Although the effect of cooling was comparatively less than ply orientation, ensuring uniform cooling across the composite thickness can effectively mitigate interlaminar thermal stresses.

The findings, including the edge effect, the stress change at the fiber plane, and influences from the ply orientations and temperature gradient, are all newly derived from a model of the PEI-matrix composite in this study. Nonetheless, the adopted simulation and analytical approach can be extended to other thermoplastic and thermosetting composites. We are confident that our insights will pave the way for enhanced designs of composite laminates subjected to intricate forming processes, as well as filament winding structures that have multiple layers with different orientations.

## Figures and Tables

**Figure 1 materials-16-06567-f001:**
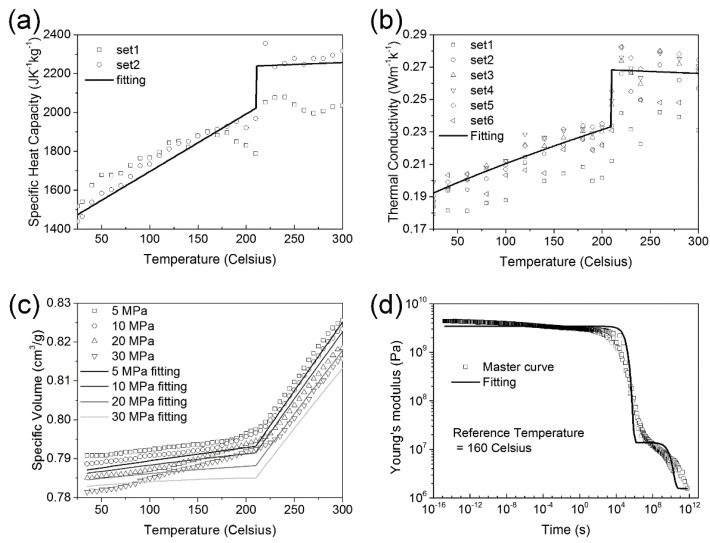
Material properties. (**a**) Specific heat capacity; (**b**) thermal conductivity; (**c**) specific volume; and (**d**) Young’s modulus. Open dots denote raw data, and solid lines represent the corresponding fitting curves [18].

**Figure 2 materials-16-06567-f002:**
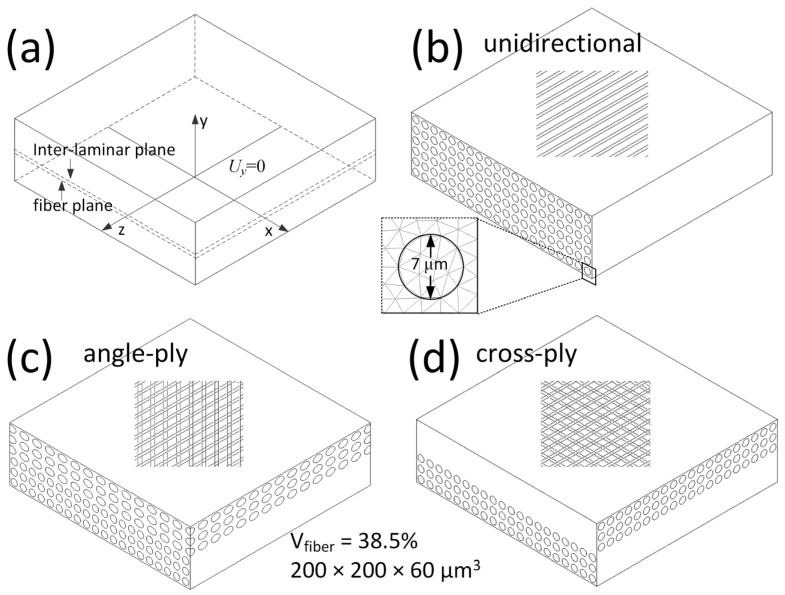
Simplified geometric models of composite laminates with different ply orientations; (**a**) model with two laminae and coordinate system; (**b**) unidirectional composite laminate; (**c**) angle-ply laminate; and (**d**) cross-ply laminate. The hatched portions in (**b**–**d**) show the transparent view of the corresponding model. The inset in (**b**) shows the mesh details.

**Figure 3 materials-16-06567-f003:**
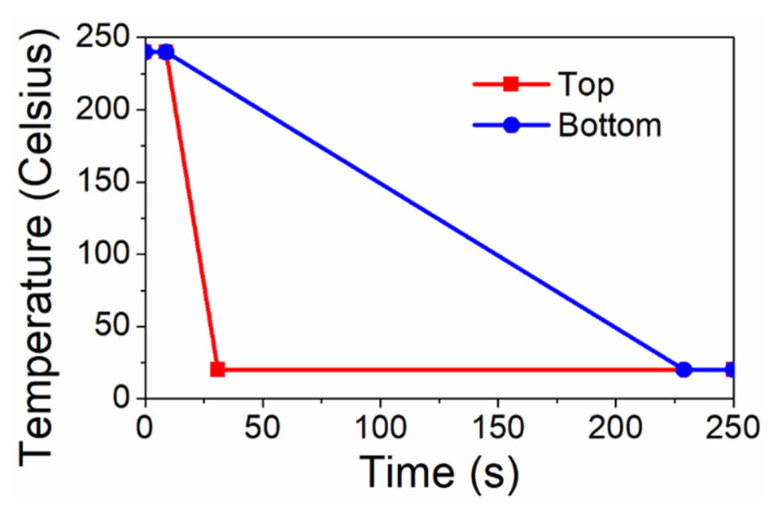
Temperature profiles used in the study.

**Figure 4 materials-16-06567-f004:**
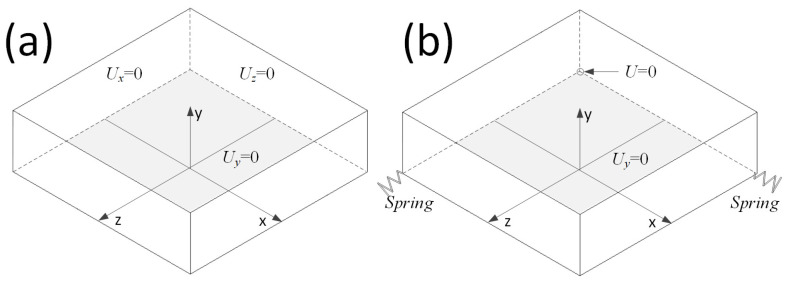
Boundary setting for (**a**) the unidirectional and cross-ply laminate models and (**b**) for the angle-ply laminate model.

**Figure 5 materials-16-06567-f005:**
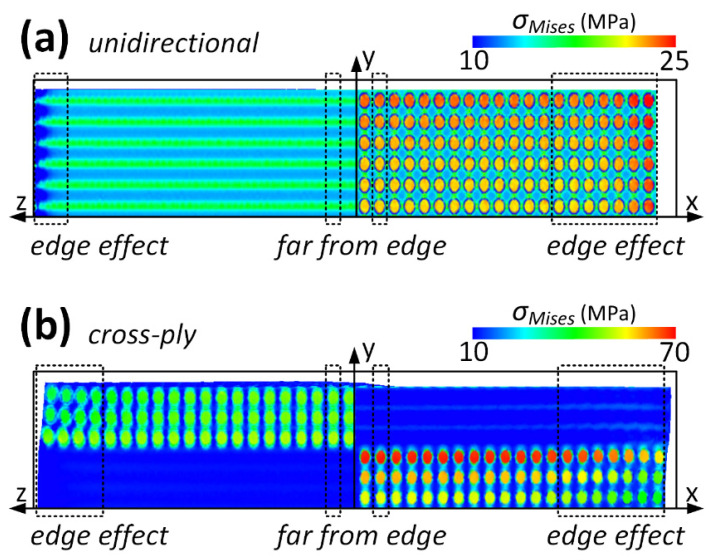
*σ_Mises_* on the cross-sections of (**a**) unidirectional and (**b**) cross-ply microscopic models under even cooling.

**Figure 6 materials-16-06567-f006:**
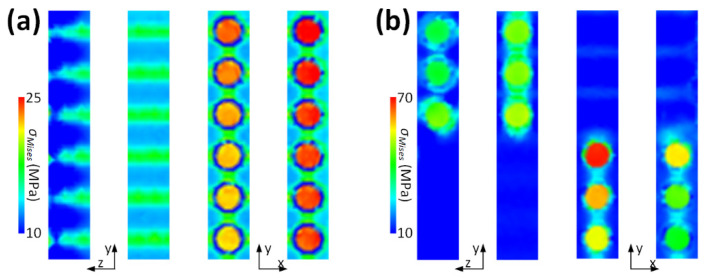
*σ_Mises_* near and far from the free edge of (**a**) unidirectional and (**b**) cross-ply laminates.

**Figure 7 materials-16-06567-f007:**
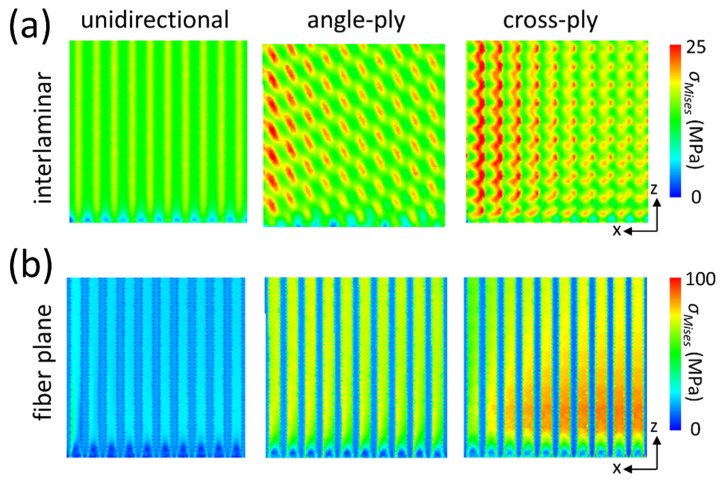
*σ_Mises_* on the (**a**) interlaminar and (**b**) fiber planes.

**Figure 8 materials-16-06567-f008:**
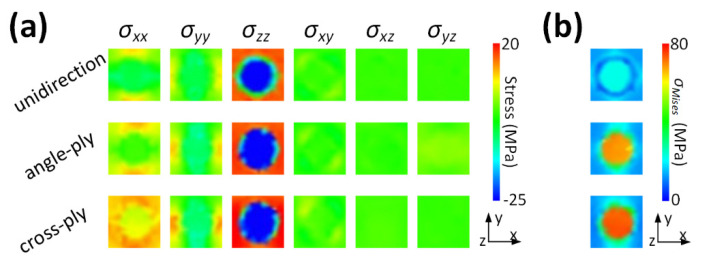
Stress distribution within a unit cell influenced by ply orientation. (**a**) Normal and shear stresses and (**b**) von Mises stress.

**Figure 9 materials-16-06567-f009:**
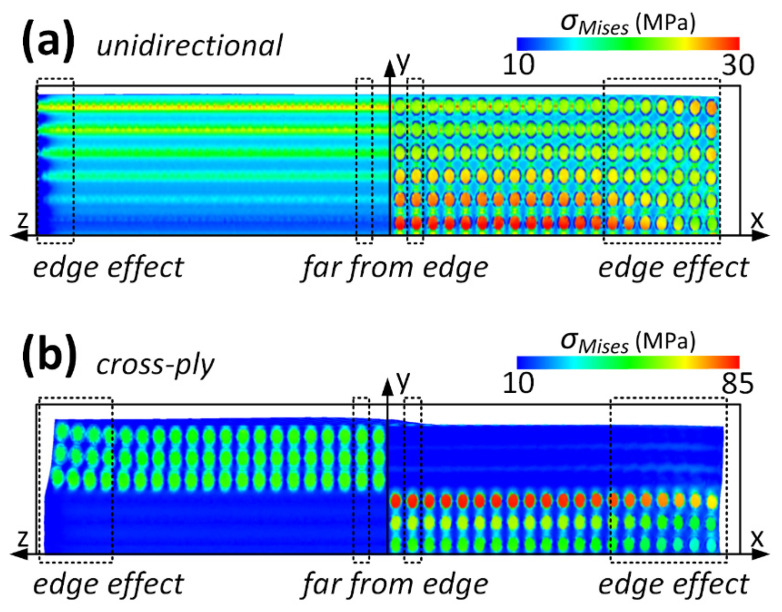
*σ_Mises_* on the cross-sections of (**a**) unidirectional and (**b**) cross-ply microscopic models under a large temperature gradient.

**Table 1 materials-16-06567-t001:** Material properties of carbon fiber.

Property	Axial	Transverse
Density (g/cm^3^)	1810	
Specific heat capacity (JK^−1^kg^−1^)	710	
Thermal conductivity (Wm^−1^K^−1^)	8.9	0.89
Coefficient of thermal expansion (−^−1^)	−4.1 × 10^−7^	7 × 10^−6^
Young’s modulus (GPa)	290	20

## Data Availability

The datasets generated during the current study are available from the corresponding author on reasonable request.

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
