# Peer review of "A Microscale Analysis of Thermal Residual Stresses in Composites with Different Ply Orientations"

_materials, 2023, doi:10.3390/ma16196567_

Round 1
Reviewer 1 Report
Title: Microscale Analysis of Thermal Residual Stresses of Composites with Different Ply Orientations
The following comments are to be considered while preparing the revised manuscript:
1. It is suggested to include the application of these kind of composite laminates (unidirectional, angle-ply and cross-ply) in the manuscript.
2. Page 2: Line 56: what does the term high-performance in-house FE software refers to? It is suggested to discuss the statement in detail for the ease in understanding. [Whitcomb et al. developed a microscale composite model with a large number of discrete fibers and resin and a high-performance in-house FE software to simulate the stresses at microscale when the composite laminate is subjected to tensile loading].
3. What are the assumptions followed in the finite element analysis for the ply orientations? It is suggested to include it for the better improvement of the manuscript.
4. Page 4: Line 145: In conditions of cooling process, whether the loading effect of the composite laminates get influenced or not?
5. How the validity and reliability of these FE were accessed? Is there any reference for this? It is suggested to provide citations for this for the better improvement of the manuscript.
6. It is noted that, the manuscript contains too many lengthy statements. It is suggested to break and reframe the sentences for better readability.
One such example is, Page 1: Line 41-44: [These conventional macroscale simulations, which define a lamina as a transversely isotropic material, ignore the composite microscopic structure consisting of fiber
and resin, resulting in an error in the stress evaluation because the microscopic stress is a few times higher than the macroscopic stress].
7. Page 3: Line 108-110: What does the terms unidirectional [0], angle-ply [0/45] and cross-ply [0/90] refers to? It is suggested to explain in detail for the ease in understanding of the readers.
8. What does the hatched portion refers to? It is suggested to represent the hatched portion in the figure 2 for the easy understanding of readers and better improvement of the manuscript.
9. Page 4: Line 122: Is there any reference for choosing the size of all the modes as 200 × 200 × 60 μm (L × W ×H)? If so, provide the citation for this.
10. It is suggested to format the references as per the journal guidelines.
Author Response
Reviewer 1:
- It is suggested to include the application of these kind of composite laminates (unidirectional, angle-ply and cross-ply) in the manuscript.
Thank you for your valuable suggestions. We have added the application of ply orientations in the revised manuscript as follows.
... be substantially enhanced. For instance, engineers often utilize ply orientations of 0, ±45, and 90° to optimize aircraft performance [1]. Modern tape placement technology has also enabled stacking laminae with alternate orientations [2]. However, this process ...
- Page 2: Line 56: what does the term high-performance in-house FE software refers to? It is suggested to discuss the statement in detail for the ease in understanding. [Whitcomb et al. developed a microscale composite model with a large number of discrete fibers and resin and a high-performance in-house FE software to simulate the stresses at microscale when the composite laminate is subjected to tensile loading].
We have carefully read the literature [14] again and found they only said “An in-house FEA code was used to perform the linear elastic analyses of this model, using the Texas A&M’s supercomputing system called Ada.” In other words, they did not describe the “in-house FE software” in more detail. Thus, we added as many descriptions as we can, as follows.
... They utilized a high-performance, in-house FE software capable of performing linear elastic analyses on a supercomputer system to simulate microscale stresses when the composite laminate undergoes tensile loading [14]. Their findings showed ...
- What are the assumptions followed in the finite element analysis for the ply orientations? It is suggested to include it for the better improvement of the manuscript.
We proposed two assumptions before conducting the FE analysis. First, it is assumed that the model is assembled from multiple square arrays which consist of carbon fiber and square matrix, because the distribution of carbon fibers is statistically uniform. Second, it is assumed that carbon fibers are solidly bonded with the surrounding matrix, because both the fiber and the PEI shrink a little during the cooling process [14],[16]. Thus, the manuscript is revised as follows.
... 38.5%, as the distribution of carbon fibers is statistically uniform. Given the minimal relative shrinkage between the carbon fiber and its surrounding matrix, it is assumed that the fibers are firmly bonded to the matrix. This implies that no friction or sliding takes place during the forming process. ...
- Page 4: Line 145: In conditions of cooling process, whether the loading effect of the composite laminates get influenced or not?
What do you mean by "loading effect"? We guess you're asking if we've taken into account the effect of pressure on the composite during the cooling process. We ignored that because we mainly focused on the impact of orientations,such as 0,±45, and 90 orientations, by controlling the control variates like the pressure.
- How the validity and reliability of these FE were accessed? Is there any reference for this? It is suggested to provide citations for this for the better improvement of the manuscript.
We have proven that the material models, mesh and software are all reliable. The temperature-dependent material models are valid through previous studies [16] and [18] as described in Section 2.1. Referring to the study on mesh [19], the element used can ensure the calculation accuracy. The result of finite element simulation is also reliable, because we have verified that the software through three different comparisons between experiments and simulations as introduced in [16,19,20].
- It is noted that, the manuscript contains too many lengthy statements. It is suggested to break and reframe the sentences for better readability.
One such example is, Page 1: Line 41-44: [These conventional macroscale simulations, which define a lamina as a transversely isotropic material, ignore the composite microscopic structure consisting of fiber
and resin, resulting in an error in the stress evaluation because the microscopic stress is a few times higher than the macroscopic stress].
We are sorry for the verbosity in our sentences. We asked a professional language editing service to polish the manuscript again. The proof is appended below.
- Page 3: Line 108-110: What does the terms unidirectional [0], angle-ply [0/45] and cross-ply [0/90] refers to? It is suggested to explain in detail for the ease in understanding of the readers.
The terms unidirectional [0], angle-ply [0/45], and cross-ply [0/90] are widely used in the composite society. Generally, the ply orientation information is filled in “[]” and different ply orientations are separated by “/”. By default, the layering starts from the tool surface and piles up layer by layer through the thickness direction. Ply orientation refers to the angle between the carbon fiber and the specified axis of a coordinate system, and the value range is from -90 to 90. [Yang, N.; Zhang, Y. Composite Aircraft Structure Design, Publisher: Aviation Industry Press, China, 2002; pp. 43-44.] We have added the definition of ply orientations in the manuscript, as follows.
... substantially enhanced. For instance, engineers often utilize ply orientations of 0, ±45, and 90° to optimize aircraft performance [1]. Modern tape placement technology has also enabled stacking laminae with alternate orientations [2]. However, this process ...
... residual stresses, we employed simulations on three representative FE models with three typical ply orientations: unidirectional [0], angle-ply [0/45], and cross-ply [0/90] laminates, under two temperature profiles. ...
... to the z-axis. Conversely, the top lamina is rotated by 0, 45, and 90° relative to the y-axis for the unidirectional [0], angle-ply [0/45], and cross-ply [0/90] laminates, respectively. The hatched ...
- What does the hatched portion refers to? It is suggested to represent the hatched portion in the figure 2 for the easy understanding of readers and better improvement of the manuscript.
The hatched portion refers to the transparent views of the representative models with layouts of unidirectional [0], angle-ply [0/45], and cross-ply [0/90] laminates. The manuscript is revised as follows:
... laminates, respectively. The hatched portions in Figure 2(b–d) provide a transparent view of the respective models, revealing the arrangement of fibers within the matrix. These configurations ...
- Page 4: Line 122: Is there any reference for choosing the size of all the modes as 200 × 200 × 60 μm (L × W ×H)? If so, provide the citation for this.
When the length is longer than the thickness, the part far from the free edge of a composite shows a uniform distribution pattern of residual stresses [14]. Therefore, the length of 200 μm is sufficient to eliminate the edge effect regarding a model with a thickness of 60 μm. We have mentioned the reason for choosing such a size in Section 3.2 “For the unidirectional and cross-ply laminates, the stress distribution within the fibers near the free edge is intricate. However, a recurring stress distribution pattern is evident across the thickness along the x or z-axis when fibers are distanced from the free edge. This edge effect also exists in the angle-ply laminate and is a pervasive phenomenon in plate-like structures where the length and width substantially exceed their thickness. A similar trend is observed in microscale models with randomly dispersed fibers, as noted in Reference [14], and in a benchmark microscale model [16].”
We have added a description of the model size in 2.2 for clarification further, as follows:
... number of DoFs. These representative FE models, having a length over three times their thickness, are capable of ...
- It is suggested to format the references as per the journal guidelines.
We have re-format the references thoroughly. For example,
Takeda, N.; Kobayashi, S.; Ogihara, S.; Kobayashi A. Experimental characterization of microscopic damage progress in quasi-isotropic CFRP laminates: effect of interlaminar-toughened layers. Adv. Compos. Mater. 1998, 7, 183–199

Reviewer 2 Report
Figure 1 (a, b), the difference in the experimental values and the (model) fitting of specific heat capacity data and thermal conductivity data is apparently very large. R2 values should have been provided. Consequently, if the model fitting values are used there will be a very big error in it.
No model equation is provided describing and linking the thermal, mechanical stresses which are used for simulating the mechanical and thermal behaviour of the different ply-oriented composites.
The results obtained are based on the models and simulation. No mathematical model/equation related to simulation is provided.
Author Response
Reviewer 2:
Comments and Suggestions for Authors:
Figure 1 (a, b), the difference in the experimental values and the (model) fitting of specific heat capacity data and thermal conductivity data is apparently very large. R2 values should have been provided. Consequently, if the model fitting values are used there will be a very big error in it.
Piecewise linear functions are commonly used to fit experimental data of both thermal conductivity and specific heat capacity [Van Krevelen, D.W.; Te Nijenhuis, K. Properties of Polymers: Their Correlation with Chemical Structure; Their Numerical Estimation and Prediction from Additive Group Contributions, 4th ed.; Elsevier: Amsterdam, The Netherlands, 2009.]. R2 for thermal conductivity and specific heat capacity is 0.934 and 0.966, respectively, indicating the sufficient accuracy of piecewise linear functions. Furthermore, the fitting results are also verified because the simulation by imposing these functions into the FE model agrees well with the experiments [18-20].
No model equation is provided describing and linking the thermal, mechanical stresses which are used for simulating the mechanical and thermal behaviour of the different ply-oriented composites.
Thank you for your invaluable suggestions. In the appendix we added in the current version of the manuscript, we describe the mathematical models of the relevant parameters in details.
Appendix
The forming process of PEI-matrix composite comprises both thermal and mechanical analyses. Equation (1) is the governing equation describing the heat transfer process, given as:
|
, |
(1) |
where is the density, is the specific heat capacity, is the thermal conductivity, is the temperature, is the time, and is the differential operator. The temperature-dependent thermal conductivity and specific heat capacity can be described by Equations (2) and (3):
|
, |
(2) |
|
, |
(3) |
where is the glass transition temperature, and are fitting parameters.
The empirical Tait equation, shown in Equation (4), can describe the specific volume, as a function of temperature and pressure:
|
, |
(4) |
where is the specific volume, is a constant, is the pressure, and are calculated from Equations (5)–(7):
|
, |
(5) |
|
, |
(6) |
|
, |
(7) |
where and denoting the liquid and solid phase, and are fitting parameters.
For an isotropic material, like PEI, its one-directional strain is one-third of the volume shrinkage. The strain can be calculated by Equation (8):
|
, |
(8) |
Equation (9) is a Maxwell model in a Prony series mathematical format, which is a typical expression of the master curve at a reference temperature.
|
, |
(9) |
where is the relaxation modulus, is the long-term modulus, and are the elastic modulus and relaxation time associated with each spring-dashpot element in the Maxwell model.
The relaxation time in Equation (9) is varied by using a suitable shift factor in Equation (10):
|
, |
(10) |
where is the shift factor, and is the reference temperature. Therefore, substituting Equation (10) into Equation (9) yields the viscoelasticity of PEI at any given temperature.
The results obtained are based on the models and simulation. No mathematical model/equation related to simulation is provided.
Thanks for your suggestions. The related mathematical models are now introduced in the appendix.

Reviewer 3 Report
Summary: This manuscript could be accepted for publication after minor revisions, according to the following comments and suggestions:
The aim of the paper, the main contributions and strengths and drawbacks:
1. The authors described three representative microscopic models consisting of discrete fiber and resin that represent unidirectional, cross-ply, and angle-ply laminates under three different cooling histories were simulated using the finite element method.
2. The manuscript is clear, well-written, relevant to the field, presented in a well-structured manner, and the data is well-organized and explained.
3. The manuscript fits the scope of the journal.
4. Minor revisions detailed below are necessary:
Introduction section
5. The references could be improved by adding at least 3 papers published in the last 2 years.
6. Novelty should be highlighted in a distinct phrase.
Materials and methods
7. Additional data regarding the software platform named “FrontCOMP_TP” and the initial parameters should be included
8. Explain more about how you choose the temperature profiles
A graphical abstract highlighting the main findings of this study could be also introduced to improve the design of this paper.
The manuscript can, in principle, be accepted after these minor revisions.
minor editing
Author Response
Reviewer 3:
Comments and Suggestions for Authors:
The references could be improved by adding at least 3 papers published in the last 2 years.
Thanks to your suggestion, we have supplemented more recently published references [1,2,20].
- Fan, S.; Zhang, J.; Wang, B.; Chen, J.; Yang. W.; Liu, W.; Li, Y. A deep learning method for fast predicting curing process-induced deformation of aeronautical composite structures. Compos. Sci. Technol. 2023, 232, 109844
- Ji, M.; Zhang. H; Wu, Q.; Zhang, H.; Huan, B.; Wang. G. Ply angle effects on cavitating flow induced structural response characteristics of the composite hydrofoils. Ocean Eng. 2023, 285, 115425
- Zhai, H.; Wu, Q.; Bai, T.; Yoshikawa, N.; Xiong, K.; Chen, C. Multi-scale finite element simulation of the thermoforming of a woven fabric glass fiber/ polyetherimide thermoplastic composite. J. Compos. Mater. 2023, 57(18) 2933–2954
Novelty should be highlighted in a distinct phrase.
Thank you for your advice. We have added a distinct phrase at the conlusion to further highlight the novelty of this study, as follows.
The findings, including the edge effect, the stress change at the fiber plane, influences from the ply orientations and temperature gradient, are all newly derived from a model of the PEI-matrix composite in this study. Nonetheless, the adopted simulation and analytical approach…
In addition, we provided a highlights herein for your information:
- Microscopic representative models with discrete fiber and resin are finite element simulated.
- Thermal residual stress has an apparent edge effect while having a repeating pattern far from the edge.
- The interlaminar residual stress suddenly change at the fiber plane near the interlaminar region.
- The normal residual stresses rather than the shear stresses are influenced by the ply orientation.
- A temperature gradient will change the thermal residual stress but has less impact compared to the ply orientation.
Additional data regarding the software platform named “FrontCOMP_TP” and the initial parameters should be included.
Thank you for your invaluable suggestions. We believe that the software platform has been introduced in Section 2.4. In the revised version, we have added more information on the software and the initial parameters, as follows.
A software platform called “FrontCOMP_TP” was employed. The software is de-signed to manage up to 100 million DoFs and operate on a high-performance computing system. This software is programmed using weak coupling of heat transfer analysis and mechanical analysis. To validate…
By leveraging a computer cluster including 576 CPU cores (Xeon E5-2660/2.20 GHz), the software platform was used to calculate the large-DoF and 250-steps (1 s = 1 step) models in Figure 2,…
Explain more about how you choose the temperature profiles.
The temperature profiles are choosen based on representativeness and simplicity. The red and blue lines can represent the fast and slow cooling, respectively, corresponding to water cooling and natural cooling. In addition, both outside-to-inside cooling and one-sided cooling can be simulated by combinging these two temperature profiles. As for the simplicity, the linear decreasing lines ensure that the composite undergoes material properties from rubbery, leathery to glassy stages. Also, ignoring the temperature over 250 °C will not affect the accumulation of residual stresses because of the low modulus and short relaxation time of PEI.
We add relevant information to the manuscript.
In Figure 3, the temperature profile is illustrated by a red line, exhibiting a decline from 240 °C at 9 s to 20 °C at a constant rate of 20 °C/s. This profile is indicative of water cooling. Even though the peak molding temperature can reach up to 350 °C, existing research indicates that minimal residual stress is accumulated above 240 °C due to PEI's low modulus and brief relaxation time [16]. However, the blue line depicts a temperature profile that cools at a rate of 2 °C/s, which is tenfold slower and represents ambient cooling. This linear decrease in temperature ideally captures the transition of PEI from its rub-bery to glassy phase, considering the glass transition temperature of PEI is roughly 210 °C. With these cooling histories in mind, we set three distinct thermal boundary conditions to probe the impact of temperature on residual stresses, assuming heat transfer is exclusive to the model's thickness direction. Firstly, the entire FE model undergoes rapid cooling, guided by the red profile, aiming to discern the basic dynamics of thermal residual stresses while eliminating the effects of the temperature gradient. Secondly, only the top surface follows the rapid-cooling profile. This simulation at-tempts to emulate real-world outside-to-inside cooling, or one-sided cooling, which naturally establishes a temperature gradient across the thickness. Finally, two Dirichlet boundaries—guided by the red and blue profiles in Figure 3—are applied to the model's top and bottom surfaces, respectively. This approach strives to recreate a cooling pro-cess with a pronounced temperature gradient, similar to that observed in water cooling.

Round 2
Reviewer 1 Report
Comments were addressed.